# Blockade of PD-L1 Enhances Cancer Immunotherapy by Regulating Dendritic Cell Maturation and Macrophage Polarization

**DOI:** 10.3390/cancers11091400

**Published:** 2019-09-19

**Authors:** Nai-Yun Sun, Yu-Li Chen, Wen-Yih Wu, Han-Wei Lin, Ying-Cheng Chiang, Chi-Fang Chang, Yi-Jou Tai, Heng-Cheng Hsu, Chi-An Chen, Wei-Zen Sun, Wen-Fang Cheng

**Affiliations:** 1Graduate Institute of Oncology, College of Medicine, National Taiwan University, Taipei 100, Taiwan; naiyunsun@gmail.com (N.-Y.S.); Handway_RC@hotmail.com (H.-W.L.); 2Department of Obstetrics and Gynecology, College of Medicine, National Taiwan University, Taipei 100, Taiwan; uly1007@yahoo.com.tw (Y.-L.C.); littlechiang1878@yahoo.com.tw (Y.-C.C.); cfangchang@gmail.com (C.-F.C.); stilabry@gmail.com (Y.-J.T.); b101092037@gmail.com (H.-C.H.); chianchen@ntu.edu.tw (C.-A.C.); 3Department of Obstetrics and Gynecology, Far Eastern Memorial Hospital, New Taipei 220, Taiwan; wenyih@mail.femh.org.tw; 4Graduate Institute of Clinical Medicine, College of Medicine, National Taiwan University, Taipei 100, Taiwan; wzsun@ntu.edu.tw; 5Department of Obstetrics and Gynecology, National Taiwan University Hospital Hsin-Chu Branch, Hsin-Chu City 300, Taiwan; 6Department of Anesthesiology, College of Medicine, National Taiwan University, Taipei 100, Taiwan

**Keywords:** antigen-specific protein vaccine, anti-PD-L1 antibody, antigen-presenting cells, dendritic cells, macrophages

## Abstract

The immuno-inhibitory checkpoint PD-L1, regulated by tumor cells and antigen-presenting cells (APCs), dampened the activation of T cells from the PD-1/PD-L1 axis. PD-L1-expressing APCs rather than tumor cells demonstrated the essential anti-tumor effects of anti-PD-L1 monotherapy in preclinical tumor models. Using the murine tumor model, we investigated whether anti-PD-L1 antibody increased the antigen-specific immune response and anti-tumor effects induced by the antigen-specific protein vaccine, as well as the possible mechanisms regarding activation of APCs. Anti-PD-L1 antibody combined with the PEK protein vaccine generated more potent E7-specific immunity (including the number and cytotoxic activity of E7-specific cytotoxic CD8^+^ T lymphocytes) and anti-tumor effects than protein vaccine alone. Anti-PD-L1 antibody enhanced the maturation of dendritic cells and the proportion of M1-like macrophages in tumor-draining lymph nodes and tumors in tumor-bearing mice treated with combinatorial therapy. PD-L1 blockade overturned the immunosuppressive status of the tumor microenvironment and then enhanced the E7 tumor-specific antigen-specific immunity and anti-tumor effects generated by an E7-specific protein vaccine through modulation of APCs in an E7-expressing small tumor model. Tumor-specific antigen (like HPV E7 antigen)-specific immunotherapy combined with APC-targeting modality by PD-L1 blockade has a high translational potential in E7-specific cancer therapy.

## 1. Introduction

Cancer cells have developed several strategies to escape from immune surveillance, such as loss of antigenicity, loss of immunogenicity, and development of an immunosuppressive microenvironment [1]. Loss of immunogenicity, including the expression of programmed death-ligand 1 (PD-L1) on tumor cells, plays a crucial role in tumor evasion of host immune responses [2]. Interaction between programmed death-1 (PD-1) receptor and its ligands PD-L1 and PD-L2 leads to inhibition of the proliferation, cytokine secretion, and cytotoxic activity of T cells [3,4]. When PD-1 expression on T cells binding with PD-L1 and/or PD-L2 expression on APCs occurs, PD-1 inhibits T cell activation by recruiting tyrosine phosphatase SHP2, which directly attenuates TCR signaling through dephosphorylation [5]. However, recent studies have indicated that PD-1-induced inhibition of T cells occurs primarily through dephosphorylation of CD28 rather than dephosphorylation of TCR by recruiting SHP2 [6]. Additionally, PD-1 is also highly expressed on Tregs, enhancing the proliferation of Tregs through interaction between PD-1 and PD-L1 [3]. Blockade of the PD-1/PD-L1 signaling pathway by anti-PD-1 or anti-PD-L1 immune checkpoint antibody therapy has produced unprecedented and durable responses in the treatment of melanoma and lung cancer patients [7]. PD-L1 expressed on tumor cells as a biomarker can predict the responses of cancer patients treated with anti-PD-L1 antibody therapy, and increased PD-L1 expression on tumor cells is associated with better anti-tumor immunity after PD-L1 blockade [8,9]. Recently, Tang et al. showed that PD-L1 expressed on APCs rather than on tumor cells might be essential to the anti-tumor efficacy of PD-L1 blockade therapy [10]. Blockade of the PD-1/PD-L1 pathway by anti-PD-1 or anti-PD-L1 Abs is considered one of the most promising modalities of cancer immunotherapy.

APCs including dendritic cells (DCs) and macrophages (Mϕs), play key roles in antigen-specific immunity. APCs can uptake tumor antigens from tumor tissues or antigen-specific cancer vaccines, migrate to secondary lymphoid organs, lymph nodes (LNs), and present antigens to T cells, leading to T cell activation [11]. Activated T cells may traffic to tumor sites for destruction. Inflammatory cytokines such as IFN-γ may induce the upregulation of PD-L1 on tumor cells [12] and APCs [13]. PD-L1 signaling may thereby inhibit T-cell functions. PD-L1 expressed on APCs may inhibit the anti-tumor immunity of T cells, and PD-L1 blockade may enhance these anti-tumor effects in vivo [10]. IFN-γ secreted by the activated T cells elicited by cancer vaccines may upregulate PD-L1 expression by tumor cells and APCs [14]. However, blockade of PD-1/PD-L1 pathways using anti-PD-1 Abs, pembrolizumab, and nivolumab produces tumor regression in only a minority of recurrent human papillomavirus (HPV)-positive oropharyngeal cancer patients [15,16,17,18]. Recently, Massarelli et al. showed that combination immunotherapy with anti-PD-1 Ab, nivolumab, and the therapeutic HPV-16 peptide vaccine inducing HPV-specific T cells could enhance anti-tumor efficacy compared with nivolumab alone for patients with incurable HPV-16-positive related cancer in a phase II clinical trial [19]. PD-L1 blockade is a potential strategy to enhance the anti-tumor effects induced by cancer vaccines.

An antigen-specific protein vaccine could generate potent antigen-specific anti-tumor effects. Our previous work demonstrated that the E7-specific fusion protein vaccine PE-E7-K3 (PEK) was able to enhance the E7-specific immunity of both CD4 and CD8 T cells by targeting the HPV-16 E7 oncoprotein, generating potent E7-specific anti-tumor effects against impalpable pulmonary tumor nodules [20]. In an attempt to improve the anti-tumor efficacy of the PEK protein vaccine, we evaluated whether combining inhibition of the PD-L1 immune checkpoint with a PEK protein vaccine could enhance anti-tumor effects in the animal model, and explored the possible mechanisms underlying the strategy. Combining the blockade of PD-L1 with the antigen-specific protein vaccine generated more potent anti-tumor activity and antigen-specific immunity compared with checkpoint blockade or vaccine alone. The blockade of PD-L1 also promoted the maturation of DCs and regulated macrophage M1-like polarization. Consequently, our results concluded that adding a PD-L1 blockade could improve the efficacy of the E7 antigen-specific protein vaccine in an E7-expressing small tumor model. This may be an innovative approach for antigen-specific immunotherapy in the future.

## 2. Results

### 2.1. Blockade of PD-L1 Enhanced the Anti-Tumor Effects of an Antigen-Specific Protein Vaccine

The therapeutic protocols used to investigate whether the blockade of PD-L1 could enhance the anti-tumor effects of an antigen-specific protein vaccine are shown in Figure 1A. Mice bearing established E7-expressing TC-1 small tumors were treated with PEK protein vaccine with or without anti-PD-L1 Ab. Anti-PD-L1 monotherapy (4605.1 ± 114.0 mm^3^) had no effect on tumor volume compared with PBS (4846.6 ± 342.1 mm^3^, *p* = 0.86, Student’s *t*-test; Figure 1B). Mice receiving PEK protein vaccine and anti-PD-L1 Ab (55.1 ± 7.1 mm^3^) had significantly smaller tumor volumes on day 28 than the group receiving PEK protein vaccine alone (348.4 ± 2.8 mm^3^, *p* < 0.001, Student’s *t*-test; Figure 1C). Moreover, 80% of the mice treated with PEK protein vaccine and anti-PD-L1 Ab were alive after 100 days of TC-1 tumor cell challenge. In contrast, none of the mice in the group receiving PEK protein vaccine with PBS survived more than 56 days after the tumor challenge (*p* < 0.001, log-rank test; Figure 1D).

### 2.2. Anti-PD-L1 Antibody Enhanced the Antigen-Specific CD8^+^ Cytotoxic and CD4^+^ Helper T Cell Immune Responses Generated by an Antigen-Specific Protein Vaccine

Representative depictions of the flow cytometric analysis for E7-specific IFN-γ-secreting CD4^+^ helper and CD8^+^ cytotoxic T lymphocytes are shown in Figure 2A. Mice vaccinated with the PEK protein vaccine and anti-PD-L1 Ab had significantly higher numbers of E7-specific IFN-γ-secreting CD4^+^ helper T precursors (147.3 ± 1.9/3.5 × 10^5^ splenocytes) than the group receiving the PEK protein vaccine alone (55.5 ± 0.5/3.5 × 10^5^ splenocytes, *p* < 0.001, Student’s *t*-test; Figure 2B). Mice receiving both the PEK protein vaccine and anti-PD-L1 Ab had also had greater numbers of E7-specific IFN-γ-secreting CD8^+^ cytotoxic T cell precursors (3599.0 ± 405.2/3.5 × 10^5^ splenocytes) than the group receiving the PEK protein vaccine alone (376.5 ± 26.2/3.5×10^5^ splenocytes, *p* = 0.002, Student’s *t*-test; Figure 2C).

### 2.3. Blockade of PD-L1 Enhanced the Tumor-Killing Activity of Antigen-Specific CD8^+^ Cytotoxic T Cells In Vitro and Ex Vivo

Representative luminescence figures of TC-1/LG tumor cells co-cultured with E7-specific CD8^+^ cytotoxic T cells with or without anti-PD-L1 Ab are shown in Figure 2D. The E7-specific CD8^+^ cytotoxic T cells treated with anti-PD-L1 Ab (8.5 ± 0.5 × 10^7^ p/s) had significantly less luminescence than those treated with isotype Ab (1.5 × 10^8^ ± 2.8 × 10^6^ p/s, *p* < 0.001, Student’s *t*-test; Figure 2E)

The ex vivo tumor killing effects of splenocytes were evaluated further. The representative luminescence activities of TC-1/LG cells co-cultured with splenocytes from mice vaccinated with the PEK protein vaccine with or without anti-PD-L1 Ab are shown in Figure 2F. The PEK protein vaccine with anti-PD-L1 Ab (7.7 ± 0.6 × 10^7^ p/s) resulted in significantly lower luminescence activity than the PEK protein vaccine alone (9.8 ± 0.2 × 10^7^ p/s, *p* = 0.017, Student’s *t*-test; Figure 2G).

### 2.4. PD-L1 is Highly Expressed in Bone Marrow Monocyte (BMM)-Derived Immature DCs, BMM-Derived M1 Macrophages, B Cells, and Tumor Cells

Representative figures of PD-L1 expression from CD3^+^ T cells (Figure 3A), natural killer (NK) cells (Figure 3B), NKT cells (Figure 3C), and CD11b^+^ Gr-1^+^ myeloid-derived suppressor cells (MDSCs) (Figure 3D) from splenocytes of mice vaccinated with the PEK protein vaccine are shown. Only low percentages of these cells expressed the PD-L1 molecule. We also investigated whether other cell types, such as BMM-derived DCs and macrophages, expressed PD-L1. As shown in Figure 3E, 11.8% of CD80^−^ CD11c^+^ DCs and 9.2% of CD86^−^ CD11c^+^ DCs expressed PD-L1. In contrast, only 4.1% of CD80^+^ CD11c^+^ DCs and 3.5% of CD86^+^ CD11c^+^ DCs expressed PD-L1 (Figure 3E). PD-L1 expression was more common among MHC-II^+^ M1-like macrophages: 87.4% of F4/80^+^ (Figure 3F) and 23.5% of MHC-II^+^ M1-like macrophages (Figure 3G) expressed PD-L1. In addition, IFN-γ induced PD-L1 expression in 83.2% of F4/80^+^ M0 macrophages (Figure 3F). Only 7.7% of CD19^+^ B lymphocytes expressed PD-L1 (Figure 3H). PD-L1 expression was not limited to immunocytes; the TC-1 tumor cells also expressed PD-L1 (Figure 3I).

### 2.5. Anti-PD-L1 Ab Enhanced the Maturation and M1-Like Polarization of Macrophages of Regional LNs

DCs and macrophages in tumor-draining lymph nodes was used to investigate whether the blockade of PD-L1 could enhance the activation of APCs in mice vaccinated with PEK protein vaccine are shown in Figure 4A. The percentages of CD80^+^ CD11c^+^ DCs (2.89 ± 0.05%) was significantly higher among mice treated with the PEK protein vaccine plus anti-PD-L1 Ab than among those treated with the PEK protein vaccine alone (2.11 ± 0.05%, *p* < 0.001, Student’s *t*-test; Figure 4B). The percentage of CD86^+^ CD11c^+^ DCs (0.91 ± 0.03%) was also significantly higher among mice treated with the PEK protein vaccine plus anti-PD-L1 Ab than among those treated with the PEK protein vaccine alone (0.63 ± 0.01%, *p* < 0.001, Student’s *t*-test; Figure 4B). In addition, the percentages of MHC-I^+^ CD11c^+^ and MHC-II^+^ CD11c^+^ DCs were significantly higher in mice treated with the PEK protein vaccine plus anti-PD-L1 antibody than among those treated with the PEK protein vaccine plus PBS (2.13 ± 0.04% vs. 1.42 ± 0.05% and 1.47 ± 0.04% vs. 1.08 ± 0.03%, respectively, *p* < 0.001 for both, Student’s *t*-test; Figure 4B).

The M1 markers, including CD80, CD86, and MHC-II molecules, of macrophages in vivo were also evaluated. The percentages of CD80^+^ F4/80^+^ and CD86^+^ F4/80^+^ macrophages were higher in mice treated with the PEK protein vaccine plus anti-PD-L1 Ab than among those treated with the PEK protein vaccine plus PBS (1.76 ± 0.09% vs. 1.55 ± 0.02%, *p* = 0.03, and 0.89 ± 0.03% vs. 0.78 ± 0.02%, *p* = 0.041, Student’s *t*-test for both; Figure 4C). In addition, the percentage of MHC-II^+^ F4/80^+^ macrophages was higher in mice treated with the PEK protein vaccine plus anti-PD-L1 Ab than among those treated with the PEK protein vaccine plus PBS (1.55 ± 0.05% vs. 1.06 ± 0.02%, *p* < 0.001, Student’s *t*-test; Figure 4C).

### 2.6. Anti-PD-L1 Ab Enhanced the Infiltration of CD4^+^ and CD8^+^ T Cells into the Tumors of Mice Vaccinated with Antigen-Specific Protein Vaccine

The protocols used to investigate whether the blockade of PD-L1 could enhance the anti-tumor immunities elicited by antigen-specific protein vaccine in tumor-infiltrating lymphocytes (TILs) are shown in Figure 5A. Representative results of the flow cytometric analysis of CD4^+^ and CD8^+^ T cells from TILs are shown in Figure 5B. The percentages of both CD4^+^ and CD8^+^ T lymphocytes were significantly higher in TILs from mice treated with the PEK protein vaccine plus anti-PD-L1 Ab than among those treated with the PEK protein vaccine plus PBS (15.42 ± 0.65% vs. 3.30 ± 0.05% and 12.14 ± 0.38% vs. 3.71 ± 0.20%, respectively, *p* = 0.002 for both, Student’s *t*-test; Figure 5C).

There was an increase in numbers of CD4^+^ and CD8^+^ T lymphocytes in the tumors of mice that were controlled by antigen-specific protein vaccine combined with blockade of PD-L1.

### 2.7. Anti-PD-L1 Ab Enhanced the Maturation of DCs and M1-Like Polarization of Macrophages in Tumor Microenvironment

Representative results of the flow cytometric analysis of tumor-infiltrating CD80^+^ CD11c^+^ and CD86^+^ CD11c^+^ DCs are shown in Figure 5D. The percentages of tumor-infiltrating CD80^+^ CD11c^+^ and CD86^+^ CD11c^+^ DCs were higher in mice treated with the PEK protein vaccine plus anti-PD-L1 Ab than among those treated with the PEK protein vaccine plus PBS (18.43 ± 0.14% vs. 11.02 ± 0.44%, *p* = 0.002, and 9.8 ± 0.12% vs. 8.32 ± 0.09%, *p* < 0.001, respectively, Student’s *t*-test for both; Figure 5E).

Regarding the M1-like polarization of tumor-infiltrating macrophages, representative results of the flow cytometric analysis of tumor-infiltrating CD80^+^ F4/80^+^ and CD86^+^ F4/80^+^ macrophages are shown in Figure 5F. The percentages of both tumor-infiltrating CD80^+^ F4/80^+^ and CD86^+^ F4/80^+^ macrophages were higher in mice treated with the PEK protein vaccine plus anti-PD-L1 Ab than among those treated with the PEK protein vaccine plus PBS (18.37 ± 0.10% vs. 13.14 ± 0.10% and 13.61 ± 0.19% vs. 9.97 ± 0.16%, respectively, *p* < 0.001 for both, Student’s *t*-test; Figure 5G).

### 2.8. Blockade of PD-L1 Promoted the Maturation of DCs In Vitro

In our previous study, immature DCs were shown to be able to express higher PD-1 and PD-L1 checkpoints than mature DCs in BMM-derived DCs and TILs, and anti-PD-1 Ab to enhance the maturation of BMM-derived DCs [21]. Next, we determined whether anti-PD-L1 Ab also enhanced the expression of various surface markers of BMM-derived DCs in vitro. Representative figures depicting the expression of CD80/86 and MHCI/II on DCs treated with anti-PD-L1 Ab are shown in Figure 6A. The anti-PD-L1 Ab generated significantly higher percentages of both CD80^+^ CD11c^+^ and CD86^+^ CD11c^+^ BMM-derived DCs than the isotype Ab (36.23 ± 0.07% vs. 25.88 ± 0.22% and 47.05 ± 0.16% vs. 25.23 ± 0.20%, respectively, *p* < 0.001 for both, Student’s *t*-test; Figure 6B). Additionally, the anti-PD-L1 Ab-treated DCs had significantly higher percentages of CD11c^+^ MHC I^+^ and CD11c^+^ MHC II^+^ cells than isotype Ab-treated DCs (47.45 ± 0.26% vs.38.77 ± 0.03% and 59.69 ± 0.20% vs. 50.26 ± 0.28%, respectively, *p* < 0.001 for both, Student’s *t*-test; Figure 6B).

### 2.9. BMM-Derived DCs Treated with Anti-PD-L1 Ab Enhanced the Activation of Antigen-Specific CD8^+^ Cytotoxic T Cells

Representative figures depicting the flow cytometric analysis for E7-specific IFN-γ-secreting CD8^+^ cytotoxic T lymphocytes are shown in Figure 6C. The number of IFN-γ-secreting E7-specific CD8^+^ T precursors was higher in mice treated with anti-PD-L1 Ab than among those treated with an isotype Ab (350.3 ± 6.7/1 × 10^5^ vs. 264.3 ± 2.0/1 × 10^5^ E7-specific CD8^+^ T cells, *p* < 0.001, Student’s *t*-test; Figure 6D).

The anti-PD-L1 Ab promoted the maturation of DCs, and these antigen-pulsed mature DCs were able to enhance the activation of antigen-specific CD8^+^ cytotoxic T precursors.

## 3. Discussion

The results of the present study revealed that PD-L1 blockade enhanced the antigen-specific anti-tumor effects and cell-mediated immunities of an antigen-specific protein vaccine beyond those of the protein vaccine alone. The anti-PD-L1 Ab increased the percentages of tumor-infiltrating CD4^+^ and CD8^+^ T cells in mice vaccinated with an antigen-specific protein vaccine. Lin et al. showed that combination therapy of anit-PD-L1 Ab and Lm-LLO-E6 vaccine which secreted HPV-16 E6 antigen fused to listeriolysin O (LLO) pore-forming toxin by bacterial vector could exhibit potent anti-tumor efficacy against HPV-16 E6-positive human TL-1 lung cancer in nude mice [22]. In comparison with Lin et al.’s study, our investigation demonstrated that anti-PD-L1 Ab could enhance antigen-specific anti-tumor effects by modulating immune cells in immune-competent mice, compared to different anti-tumor mechanisms through neutralizing E6 protein by Lm-LLO-E6 vaccine to further inhibit PD-L1 expression, and consequently could suppress tumor growth in immunocompromised mice [22]. Recently, increasing evidence has emerged in support of the hypothesis that the anti-tumor mechanisms of anti-PD-L1 Ab in cancer immunotherapy come mainly through its targeting of PD-L1-expressed antigen-presenting cells, such as DCs and macrophages [10,23]. In the current study, the anti-PD-L1 Ab also enhanced the maturation of DCs and M1-like polarization of macrophages in both regional LNs and tumors in mice. In addition, anti-PD-L1 Ab enhanced the maturation of BMM-derived DCs in vitro. The blockade of PD-L1 enhanced the anti-tumor efficacy of an E7 antigen-specific protein vaccine by modulating the maturation of DCs and M1-like polarization in macrophages in an E7-expressing small tumor model.

DCs play a central role in antigen-specific anti-tumor T cell immune responses. The maturation status of DCs determines their potency during antigen presentation. Immature DCs that reside in nonlymphoid tissues are able to engage in antigen uptake and processing, but do not provide the signals needed to prime T cell responses [24]. Mature DCs induced by extraneous signals are able to migrate to secondary lymphoid organs to upregulate immunogenicity to initiate T cell responses and downregulate antigen capture and antigen-processing abilities [24,25]. Additionally, tumor cells are able to reverse these mature, tumor-infiltrating, immune-active anti-tumor DCs into immune-suppressive, pro-tumor DCs by upregulating inhibitory receptors, including PD-1 and T cell immunoglobulin and mucin domain 3 (TIM-3) molecules on DCs [26].

Macrophages tend to be polarized toward an M2-like phenotype associated with repair and immune suppression (pro-tumor), rather than the M1-like phenotype associated with inflammation and immunity (anti-tumor) in established tumors [27,28]. Interleukin (IL)-10 secreted by tumor-associated M2-like macrophages inhibits CD8^+^ T cell responses to chemotherapy by suppressing the inflammatory cytokine IL-12 from tumor-infiltrating DCs [29]. In addition, arginase-1 induction and PD-L1 expression by tumor-associated M2-like macrophages can suppress T cell immune responses [30,31]. Therefore, identifying ways to promote the maturation of DCs and M1 shifting of macrophages has became an important strategy for cancer immunotherapy.

Our previous study demonstrated that the PEK protein vaccine is able to generate potent antigen-specific immunity, including antigen-specific CD8^+^ and CD4^+^ T cell precursors, antigen-specific Abs, and anti-tumor effects to control micro-metastatic tumor lesions in a pulmonary metastatic model [20]. However, the PEK protein vaccine alone is not potent enough to control established tumors in a subcutaneous tumor model (Figure 1C,D). Therefore, a strategy is required to enhance the anti-tumor effects of the antigen-specific protein vaccine.

In addition to tumor cells, PD-L1 is upregulated on APCs, including DCs and macrophages, in tumor microenvironments due to inflammatory stimuli [32]. Tumor cells or APCs expressing PD-L1 inhibitory ligand can downregulate the amplitude of T cell activation and suppress anti-tumor immunity through the PD-1/PD-L1 axis [10,33]. Moreover, in addition to PD-L1-expressing tumor cells, PD-L1-expressing tumor-infiltrating myeloid cells appeared to provide a compensatory source of the inhibitory ligand to downregulate the anti-tumor immune responses mediated by T cells in mice [34]. Lin et al. further demonstrated that PD-L1 expressed on host immune cells, including DCs and macrophages, rather than PD-L1 intrinsically expressed on tumor cells could account for the anti-tumor efficacy of anti-PD-L1 Ab monotherapy [23]. Lin et al. first demonstrated that E7-specific Sig/E7/LAMP-1 vaccinia could generate in vivo protection against TC-1 tumor [35]. They also demonstrated that treatment with the Sig/E7/LAMP-1 vaccinia vaccine could only cure mice with small TC-1 tumors [35]. Our study revealed that PD-L1 blockade could enhance the anti-tumor effects of E7-sepcific PEK protein vaccine against established TC-1 subcutaneous tumors. PD-L1 was also highly expressed on APCs, including immature DCs, M1 macrophages, IFN-γ-induced M0 macrophages, and B cells in our study (Figure 3E–H), which implies that the blockade of the PD-1/PD-L1 axis in these cells might be a potential strategy to enhance anti-tumor immunity by blocking the inhibitory signaling pathway.

Antibody-mediated blockade of PD-L1 has shown an induced unprecedented durable response in patients with advanced cancers [36]. Additionally, PD-L1 expressed in APCs rather than on tumor cells could play an essential role in PD-L1 blockade therapy [10]. However, the majority of cancers present resistance to PD-L1 blockade monotherapy due to a lack of pre-existing tumor-infiltrating anti-tumor CD8^+^ T cells [37,38]. Cancer vaccines could allow the priming and intratumoral recruitment of CD8^+^ T cells and transform a “non-inflamed” non-permissive tumor resistant to checkpoint blockade into a sensitive “inflamed” tumor [39]. Our survey revealed that the combination of an antigen-specific protein vaccine with anti-PD-L1 therapy increased the number of tumor-infiltrating CD4^+^ and CD8^+^ T cells within tumors (Figure 5B,C). Compared with our anti-PD-L1 primary resistance tumor model, innate immune agonists such as TLR agonists, the Sting pathway, and oncolytic viruses might enhance the anti-tumor effect treated with anti-PD-L1 Ab in PD-L1 sensitive tumor models.

PD-L1 expression on DCs can attenuate the anti-tumor effects of effector T cells by interacting with PD-1 on the surface of T lymphocytes to induce T-cell apoptosis, anergy, and exhaustion [40]. Blockade of PD-L1 has been shown to promote the maturation and function of DCs in the peripheral blood of patients with colorectal cancer in vitro [41]. We determined that anti-PD-L1 Ab was able to enhance DC maturation (Figure 6A,B) and activate antigen-specific cytotoxic CD8^+^ T cells (Figure 6C,D). The results of the in vitro experiments were demonstrated further in the tumor microenvironment in vivo (Figure 5D,E). Recently, Lin et al. also showed that the PD-L1 expression of intratumoral macrophages rather than that of tumor cells could account for the potential anti-tumor efficacy of PD-L1 blockade therapy [23]. We also demonstrated that anti-PD-L1 Ab improved the proportion of M1 in macrophages of draining LNs (Figure 4C) and tumor microenvironments (Figure 5F,G).

## 4. Materials and Methods

### 4.1. Cell Lines

The TC-1 tumor cells (ExPASy Cellosaurus database, Mouse lung) (RRID: CVCL_4699) [35] and E7-specific CD8^+^ T cell line [42] were produced and maintained as described previously. The E7-specific CD8^+^ T cell line was a kind gift from Dr. T-C Wu (Johns Hopkins Medical Institution, Baltimore, MD, USA), and was generated from splenocytes of E7-vaccinated C57BL/6 mice and co-cultured with irradiated TC-1 tumor cells and 10 IU interleukin (IL)-2 weekly [43]. Luciferase/GFP-expressing TC-1 (TC-1/LG) cells were generated by transducing TC-1 cells with lentiviral vector [44]. Briefly, 293T cells generated lentiviral particles through transfection of pCMVΔR8.91, pMDG (Academia Sinica, Taipai, Taiwan), and pLKO/luciferase/AS3.1.EGFP3, in which the luciferase gene was subcloned from pGL2-basic (Promega, Madison, WS, USA) into pLKO/AS3.1.EGFP3 (Academia Sinica, Taipei, Taiwan). TC-1 cells were then infected with lentivirus using 8 μg/mL polybrene (Sigma, St Louis, MO, USA). The TC-1/LG cells were sorted by flow cytometry, cultured, and then used in subsequent experiments. All experiments were performed with mycoplasma-free cells.

### 4.2. Preparation and Vaccination of Protein Vaccines

The E7-specific protein vaccine PE-E7-K3 (PEK) (600 μg/vial), and the adjuvant GPI-0100 (600 μL/vial; 1 mg/mL) were kindly provided by Vax Genetics Vaccine Co., LTD (HsinChu County, Taiwan). The PEK vaccine was mixed with the GPI-0100 (1:1) and incubated for 16 h at 4 °C. For the in vivo anti-tumor experiments, the mixed PEK/GPI-0100 protein vaccine (100 μL) was subcutaneously injected into the bilateral axillary and inguinal areas of the mice.

### 4.3. Mice

Female C57BL/6 mice, 6 to 8 weeks old, were purchased from the National Laboratory Animal Center (Taipei, Taiwan) and bred in the animal facility of the School of Medicine of National Taiwan University. All of the animal procedures were conducted according to approved protocols and in accordance with recommendations for the proper use and care of laboratory animals. (Animal handling and procedures were approved by the animal ethic committee of College of Medicine, National Taiwan University. Ethic Code: “20140053” and “20190074”).

### 4.4. Administration of Anti-PD-L1 Ab

Rat anti-mouse PD-L1 (clone 10F.9G2) Ab and rat IgG2b isotype Ab (clone LTF-2) were purchased from BioXCell (West Lebanon, NH, USA). The anti-PD-L1 Ab was diluted with PBS and 200 μg was intraperitoneally injected into each mouse at the indicated intervals.

### 4.5. In Vivo Tumor Treatment Experiments with Protein Vaccine and/or Anti-PD-L1 Ab

The treatment protocols for PEK protein vaccine and/or anti-PD-L1 Ab are presented in Figure 1A. Tumor diameter was measured using calipers weekly starting 7 days after tumor challenge; the tumor volume was defined by 4πR^3^/3, where R is the radius of the tumor. Mice were euthanized when their tumor reached 2 cm in diameter or when they appeared sick; this was recorded as death for the survival curve.

### 4.6. Intracellular IFN-γ Staining and Flow Cytometry

Mice were challenged with TC-1 tumor cells and immunized with the PEK protein vaccine and/or anti-PD-L1 Ab (Figure 1A). Splenocytes were harvested 7 days after the last protein immunization (=on day 28 after tumor challenge) and incubated with 1 μg/mL of MHC I-restricted E7 peptide (aa49–57) or 10 μg/mL of MHC II-restricted E7 peptide (aa30–67) overnight. Golgistop (BD Biosciences, San Jose, CA, USA ) was added 6 h before harvesting. The harvested splenocytes were stained with PE-conjugated anti-CD4 or anti-CD8a Ab (BD Pharmingen), and then fixed and permeabilized using the Cytofix/Cytoperm kit (BD Pharmingen). The splenocytes were then stained with fluorescein isothiocyanate (FITC)-conjugated anti-IFN-γ Ab (BioLegend, San Diego, CA, USA) and analyzed using a FACSCalibur flow cytometer (BD Biosciences).

### 4.7. In Vitro and Ex Vivo Tumor-Killing Activity

E7-specific CD8^+^ T cells were cultured with irradiated TC-1/LG tumor cells (1:4 ratio) in a 96-well plate (2 × 10^4^ cells/well) for 24 h in the presence of anti-PD-L1 Ab or isotype Ab (10 μg/mL). Luciferin (Promega) was added and the total flux (p/s) from each well was measured using IVIS Imaging Systems (Caliper Life Sciences, Alameda, CA, USA).

Splenocytes were harvested from various groups (=on day 28 after tumor challenge) as described earlier, and then co-cultured with irradiated TC-1/LG cells (10:1 ratio) in a 96-well plate (2 × 10^4^ cells/well) for 24 h, and luciferin was added to detect the total flux (p/s) from each well, as described earlier.

### 4.8. Preparation of BMM-Derived DCs and Macrophages

Bone marrow monocyte (BMM)-derived dendritic cells (DCs) [45] or macrophages [46] from mice were prepared as described previously, with some modifications. Bone marrow cells were collected from the femurs and tibias by flushing, and the cells were filtered through a 70 μm cell strainer (BD Falcon, San Jose, CA, USA). Red blood cells were removed using RBC lysis buffer (eBioscience, San Diego, CA, USA).

For BMM-derived DCs, bone marrow cells (1 × 10^6^ cells/well) were cultured in 24-well plates in CTL culture medium containing 3% FBS supplemented with 10^3^ U/mL recombinant murine granulocyte–macrophage colony-stimulating factor (GM-CSF; PeproTech, Rocky Hill, NJ, USA) in a 5% CO_2_ atmosphere at 37 °C for 6 days. Fresh GM-CSF-containing medium was replaced every 2 days. Immature DCs were treated with lipopolysaccharide (LPS, 50 ng/mL; Sigma-Aldrich Chemie GmbH, Taufkirchen, Germany) at day 6 overnight to generate mature DCs.

For BMM-derived macrophages, bone marrow cells (2 × 10^6^ cells/well) were cultured in 6-well plates in RPMI 1640 culture medium containing 10% FBS supplemented with 10^3^ U/mL recombinant murine macrophage colony-stimulating factor (M-CSF; PeproTech, Rocky Hill, CT, USA) in a 5% CO_2_ atmosphere at 37 °C for 6 days. To generate M0 macrophages, BMM-derived macrophages were harvested on day 7. To generate M1 macrophages, BMM-derived macrophages were treated with 50 ng/mL LPS and 10^3^ U/mL GM-CSF (PeproTech) in a 5% CO_2_ atmosphere at 37 °C overnight.

### 4.9. PD-L1 Expression of Immune Cells and Tumor Cells Analyzed by Flow Cytometric Analysis

To determine the expression of PD-L1 by various immune cells and mature/immature BMM-derived DCs or M0/M1 BMM-derived macrophages, these cells were obtained, cultured, and harvested as described previously. To evaluate the effect of IFN-γ on PD-L1 expression of macrophages, BMM-derived M0 macrophages were treated with 500 U/mL recombinant murine IFN-γ (PeproTech) in a 5% CO_2_ atmosphere at 37 °C for 24 h. To visualize surface PD-L1 expression on T cells, NK cells, NKT cells, MDSCs, and B cells, splenocytes were stained with FITC-conjugated anti-CD3 Ab (BioLegend), PE-conjugated anti-NK1.1 (BioLegend), PE-conjugated anti-CD11b (BioLegend), FITC-conjugated anti-Gr-1 (BioLegend), PE-conjugated anti-CD19 (BioLegend), and PerCP-eFluor 710-conjugated anti-CD274 (PD-L1) (eBioscience), respectively. To visualize the surface expression of PD-L1 on DCs and macrophages, BMM-derived DCs or BMM-derived macrophages were stained with PE-conjugated anti-CD11c Abor anti-F4/80 Ab (BioLegend), FITC-conjugated anti-PD-L1 Ab (BioLegend), and PE-Cy5-conjugated anti-CD80 Ab (BioLegend) or PE-Cy5-conjugated anti-CD86 or anti-MHC-II Ab (BioLegend), respectively. To visualize surface PD-L1 expression on TC-1 tumor cells, cells were stained with PE-conjugated anti-PD-L1 Ab (BioLegend). Flow cytometric analysis was performed using a FACSCalibur flow cytometer with CELLQuest software (BD Biosciences).

### 4.10. In Vivo Maturation Status of DCs and M1-Like Polarization of Macrophages from Regional LNs

Tumor-draining LNs were harvested from mice treated with E7-specific protein vaccine and/or anti-PD-L1 Ab 7 days after the last protein vaccination (on day 28 after tumor challenge) (Figure 4A). Single cell suspensions were prepared as described previously [47]. To detect the maturation status of DCs and the M1-like polarization of macrophages, the cells were stained with FITC-conjugated anti-CD11c Ab, FITC-conjugated anti-F4/80 Ab, PE-conjugated anti-CD80 Ab, and PE-Cy5-conjugated anti-CD86 Ab (BioLegend) or PE-conjugated anti-MHC class I and PE-Cy5-conjugated anti-MHC class II Abs (BioLegend). The cells were analyzed by flow cytometry, as described above.

### 4.11. Isolation of Tumor-Infiltrating Lymphocytes (TILs)

TILs were prepared as described previously, with some modifications [47]. Briefly, mice were challenged with TC-1 tumor cells and immunized with PEK protein vaccine and/or anti-PD-L1 Ab. The mice were sacrificed and tumors excised 7 days after the last protein immunization (on day 28 after tumor challenge) (Figure 5A). The tumors were dissected into small fragments and digested in 0.1 mg/mL collagenase in CTL medium at 37 °C overnight. After filtering through a 40 μm cell strainer (BD Falcon), the cell suspension was incubated for 30 min at 37 °C. After washing with CTL medium, mixing cell suspensions of CTL medium and balanced salt medium were layered on Ficoll–Paque medium (GE Healthcare, Pittsburgh, PA, USA) before centrifugation. TILs from the white interface layer were collected and washed with PBS.

### 4.12. Detection of CD4^+^ T Lymphocytes, CD8^+^ T Lymphocytes, DCs, and Macrophages from TILs

CD4^+^ T cells and CD8^+^ T cells from the prepared TILs were stained with FITC-conjugated anti-CD3 Ab (BD Pharmingen), PE-conjugated anti-CD4 Ab, and PE-conjugated anti-CD8 Ab (BD Pharmingen). DCs from the prepared TILs were stained with FITC-conjugated anti-CD11c Ab (BD Pharmingen), PE-conjugated anti-CD80Ab (BD Pharmingen), and PE-Cy5-conjugated anti-CD86Ab (BioLegend) as described previously [48]. Macrophages from the prepared TILs were stained with FITC-conjugated anti-F4/80 Ab (BD Pharmingen), PE-conjugated anti-CD80Ab (BD Pharmingen), and PE-Cy5-conjugated anti-CD86Ab (BioLegend). The cells were analyzed by flow cytometry, as described above.

### 4.13. Maturation Status of BMM-Derived DCs Treated with Anti-PD-L1 Ab

To determine whether anti-PD-L1 Ab enhanced DC maturation, BMM-derived DCs treated with 50 ng/mL LPS were used as a positive control. DCs were treated with or without LPS and anti-PD-L1 Ab (50 μg/mL) in a 5% CO_2_ atmosphere at 37 °C overnight. The cells were stained with FITC-conjugated anti-CD11c Ab (BioLegend), PE-conjugated anti-CD80 Ab (BioLegend), anti-MHC class I Ab (Biolegend), PE-Cy5-conjugated anti-CD86 Ab (BioLegend) or anti-MHC class II Ab (Biolegend). The cells were analyzed by flow cytometry, as described above.

### 4.14. BMM-Derived DCs Treated with Anti-PD-L1 Ab Stimulated the Activation of Antigen-Specific Cytotoxic CD8^+^ T Cells

The BMM-derived DCs (1 × 10^6^ cells/well) were pulsed with 10 μg/mL of MHC I-restricted E7 peptide (aa49–57) at day 7 for 6 h, and then treated with anti-PD-L1 Ab (50 μg/mL) or isotype Ab (50 μg/mL) overnight, and co-cultured with the E7-specific CD8^+^ T cell line (1:10 ratio) at day 8 overnight. Golgistop (BD Pharmingen) was added 6 h before harvesting the cells. The co-cultured cells were first stained with allophycocyanin-conjugated anti-CD3 Ab (BioLegend) and PerCP-Cy5.5 conjugated anti-CD8a Ab (BD Pharmingen), then fixed and permeabilized using the Cytofix/Cytoperm kit (BD Pharmingen). The cells were then stained with FITC-conjugated anti-IFN-γ Ab (BioLegend) and analyzed by flow cytometry, as described above.

### 4.15. Statistical Analysis

All of the data were expressed as mean ± SEM (standard error), which represented at least three different experiments. Student’s t-test was used to detect statistical significance. The log-rank test was used to evaluate data from survival experiments. All *p* values < 0.05 were considered statistically significant.

## 5. Conclusions

The strategy of PD-L1 blockade had the effect of increasing the potency of the anti-tumor effects of the E7 antigen-specific protein vaccine, which could directly increase the susceptibility of tumor cells to killing and indirectly enhance the antigen-specific CD8^+^ T-cell responses through promoting DC maturation and M1-like polarization of macrophages to overturn the immuno-suppressive tumor microenvironment in an E7-expressing tumor model. Therefore, tumor specific antigen (like HPV E7 antigen)-specific immunotherapy combined with APCs targeting modality by PD-L1 blockade has high translational potential in E7-specific cancer therapy.

## Figures and Tables

**Figure 1 cancers-11-01400-f001:**
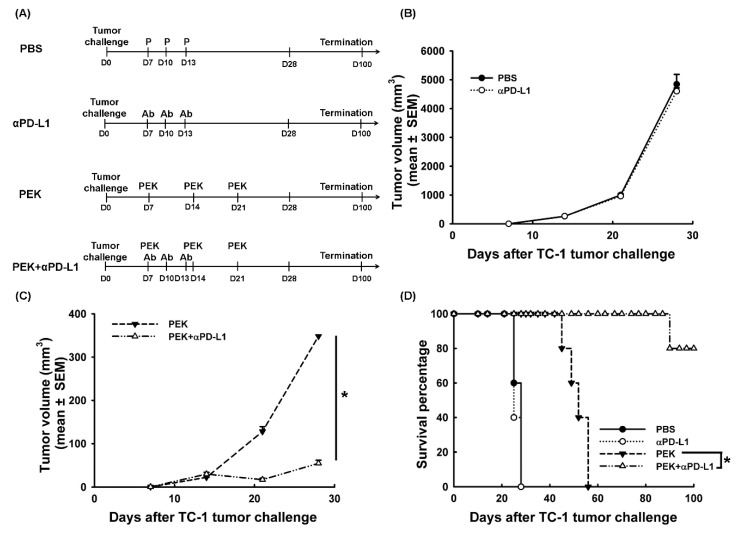
Anti-tumor effects of PEK antigen-specific protein vaccine with or without anti-PD-L1 Ab. (**A**) Diagram of the different regimens of PEK protein vaccine with or without anti-PD-L1 Ab. (**B**) Anti-tumor effects of mice treated with PBS or anti-PD-L1 Ab alone. Anti-tumor effects were not improved in tumor-bearing mice treated with anti-PD-L1 Ab alone (*p* = 0.86, Student’s *t*-test). (**C**) Anti-tumor effects of mice treated with PEK protein vaccine with or without anti-PD-L1 Ab. Tumor volumes were significantly lower in mice treated with the PEK protein vaccine plus anti-PD-L1 Ab than in mice treated with the PEK protein vaccine plus PBS (*: *p* < 0.001, Student’s *t*-test). (**D**) Overall survival of mice treated with PEK protein vaccine with or without the anti-PD-L1 Ab. Eighty percent of mice that received PEK protein vaccination combined with anti-PD-L1 Ab were still alive 100 days after TC-1 tumor challenge. None of the mice receiving the PEK protein vaccine plus PBS survived more than 60 days of tumor challenge (*: *p* < 0.001, log-rank test). All experiments were performed independently in triplicate (mean ± SEM).

**Figure 2 cancers-11-01400-f002:**
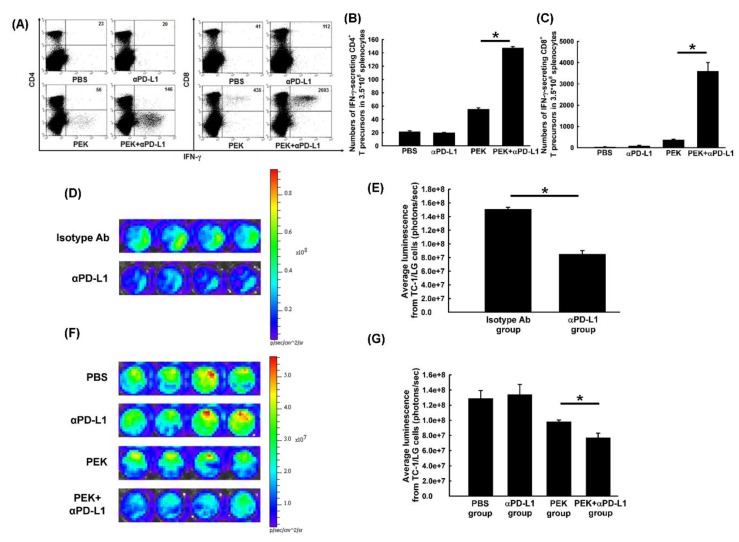
Antigen-specific immunoprofiles of mice treated with PEK protein vaccine with or without anti-PD-L1 Ab, and in vitro or ex vivo tumor-specific killing activities of antigen-specific CD8^+^ cytotoxic T cells or splenocytes from various immunized groups. (**A**) Representative figures of E7-specific IFN-γ-secreting CD4^+^ helper or CD8^+^ cytotoxic T cell precursors/3.5 × 10^5^ splenocytes on day 28 after tumor challenge in various vaccinated groups, as determined by flow cytometry analysis. (**B**) Bar figures depict the number of E7-specific IFN-γ-secreting CD4^+^ helper T cell precursors/3.5 × 10^5^ splenocytes on day 28 after tumor challenge in various groups, as determined by flow cytometry (*n* = 5 per group, mean ± SEM). Mice treated with the PEK protein vaccine combined with anti-PD-L1 Ab had significantly higher numbers of E7-specific CD4^+^ T cell precursors than the other groups (*: *p* < 0.001, Student’s *t*-test). (**C**) Bar figures depict the number of E7-specific IFN-γ-secreting CD8^+^ cytotoxic T cell precursors/3.5 × 10^5^ splenocytes on day 28 after tumor challenge in various groups, as detected by flow cytometry (*n* = 5 per group, mean ± SEM). The number of E7-specific CD8^+^ T cell precursors was highest in mice treated with the PEK protein vaccine combined with anti-PD-L1 Ab (*: *p* = 0.002, Student’s *t*-test). All experiments were performed independently in triplicate. (**D**) Representative figures of the in vitro tumor-specific killing activities of E7-specific CD8^+^ cytotoxic T cells treated with or without anti-PD-L1 Ab. (**E**) Quantitation of the average luminescence of TC-1-LG cells from mice in various treatment groups. The T cells from mice treated with anti-PD-L1 Ab exhibited lower luminescence than those from mice not treated with anti-PD-L1 Ab (*: *p* < 0.001, Student’s *t*-test). (**F**) Representative figures of the ex vivo tumor-specific killing activities of splenocytes from mice on day 28 after tumor challenge in various groups. (**G**) Quantitation of the average luminescence of TC-1-LG cells co-cultured with splenocytes from mice on day 28 after tumor challenge in various treatment groups (*n* = 5 per group, mean ± SEM). Splenocytes from mice treated with PEK protein vaccine plus anti-PD-L1 Ab exhibited lower luminescence than splenocytes from mice treated with the PEK protein vaccine plus PBS (*: *p* = 0.017, Student’s *t*-test). All experiments were performed independently in triplicate.

**Figure 3 cancers-11-01400-f003:**
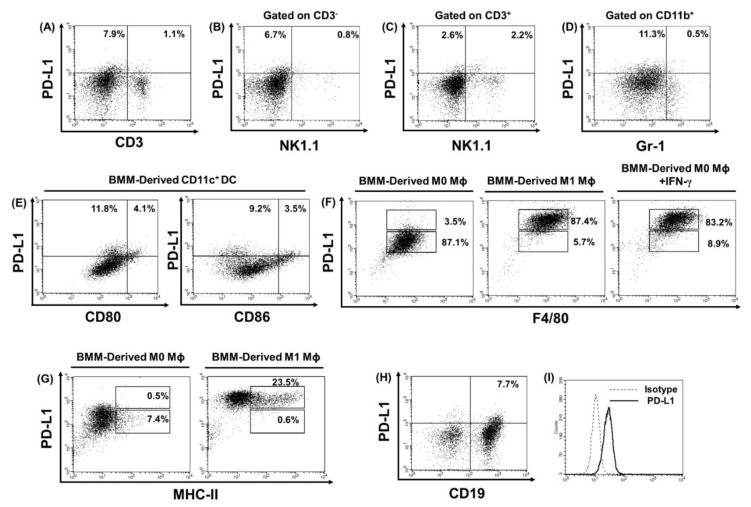
PD-L1 expression on various immunocytes from splenocytes of mice treated with PEK protein vaccine and bone marrow-derived cells, and TC-1 tumor cells analyzed by flow cytometry. (**A**) Representative figures depict PD-L1 expression on T lymphocytes. (**B**) PD-L1 expression on natural killer (NK) cells of splenocytes. (**C**) PD-L1 expression on natural killer T (NKT) cells in splenocytes. (**D**) PD-L1 expression on CD11b^+^ Gr-1^+^ myeloid-derived suppressor cells (MDSCs) from splenocytes. Only low percentages of T cells, NK cells, NKT cells, and MDSCs from splenocytes expressed PD-L1. (**E**) PD-L1 expression on bone marrow monocyte (BMM)-derived CD11c^+^ DCs. Greater percentages of BMM-derived immature CD80^−^ (11.8%) or CD86^−^ (9.2%) CD11c^+^ DCs expressed PD-L1 compared with BMM-derived mature CD80^+^ (4.1%) or CD86^+^ (3.5%) CD11c^+^ DCs. (**F**) PD-L1 expression on BMM-derived M0, IFN-γ-treated M0-, and M1-like F4/80^+^ macrophages. (**G**) PD-L1 expression on BMM-derived M0- and M1-like MHC-II^+^ macrophages. (**H**) PD-L1 expression on B cells of splenocytes. (**I**) PD-L1 expression on TC-1 tumor cells. All experiments were performed independently in triplicate.

**Figure 4 cancers-11-01400-f004:**
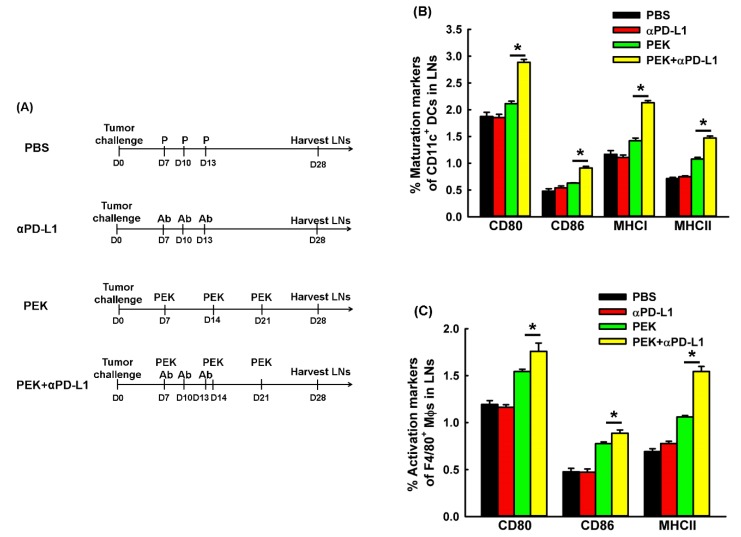
The effects of PD-L1 inhibition on the maturation of DCs and M1-like polarization of macrophages in tumor-draining lymph nodes (LNs) from mice vaccinated with the PEK protein vaccine. (**A**) Schematic graph showing the harvested processes of in vivo DCs and macrophages from tumor-draining LNs in mice vaccinated with PEK protein vaccine with or without anti-PD-L1 Ab. (**B**) Percentages of maturation status of DCs in tumor-draining lymph nodes from mice on day 28 after tumor challenge in various groups, as determined by flow cytometry (*n* = 5 per group, mean ± SEM). The percentage of CD11c^+^ DCs expressing CD80 significantly increased in mice treated with the PEK protein vaccine plus anti-PD-L1 Ab compared with mice treated with the PEK vaccine plus PBS (*: *p* < 0.001, Student’s *t*-test). The percentage of CD11c^+^ DCs expressing CD86 significantly increased in mice treated with the PEK vaccine plus anti-PD-L1 Ab compared with mice treated with the PEK vaccine plus PBS (*: *p* < 0.001, Student’s *t*-test). The percentage of MHC-I-expressing CD11c^+^ DCs was significantly higher in mice treated with the PEK vaccine plus anti-PD-L1 Ab than among those treated with the PEK vaccine plus PBS (*: *p* < 0.001, Student’s *t*-test). The percentage of MHC-II-expressing CD11c^+^ cells was significantly higher in mice treated with the PEK vaccine plus anti-PD-L1 Ab than among those treated with the PEK vaccine plus PBS (*: *p* < 0.001, Student’s *t*-test). (**C**) Percentages of activation markers of F4/80^+^ macrophages in tumor-draining lymph nodes from mice on day 28 after tumor challenge in various groups, as determined by flow cytometry (*n* = 5 per group, mean ± SEM). The percentage of CD80-expressing F4/80^+^ macrophages was significantly higher in mice treated with the PEK vaccine plus anti-PD-L1 Ab than among those treated with the PEK vaccine plus PBS (*: *p* = 0.03, Student’s *t*-test). The percentage of CD86-expressing F4/80^+^ macrophages was significantly higher in mice treated with the PEK vaccine plus anti-PD-L1 Ab than among those treated with the PEK vaccine plus PBS (*: *p* = 0.041, Student’s *t*-test). The percentage of MHC-II-expressing F4/80^+^ macrophages was significantly higher in mice treated with the PEK vaccine plus anti-PD-L1 Ab than among those treated with the PEK vaccine plus PBS (*: *p* < 0.001, Student’s *t*-test). All experiments were performed independently in triplicate.

**Figure 5 cancers-11-01400-f005:**
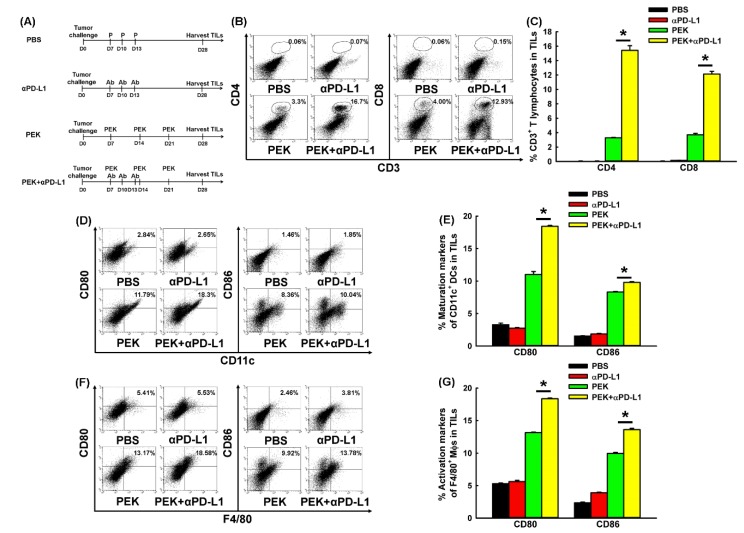
Alteration of T cells, DCs, and macrophages from tumor-infiltrating lymphocytes (TILs) of mice treated with PEK protein vaccine and/or anti-PD-L1 Ab. (**A**) Schematic diagram showed the obtained TILs from mice receving PEK protein vaccine with or without anti-PD-L1 Ab. (**B**) Representative figures depicting tumor-infiltrating CD4^+^ and CD8^+^ T lymphocytes from mice on day 28 after tumor challenge in various groups, as determined by flow cytometry. (**C**) Percentages of CD4^+^ and CD8^+^ T lymphocytes from mice on day 28 after tumor challenge in various groups, as determined by flow cytometry (*n* = 5 per group, mean ± SEM). The percentage of CD4^+^ T lymphocytes was higher in tumors from mice treated with PEK protein vaccine plus anti-PD-L1 Ab than in tumors from mice treated with PEK vaccine plus PBS (*: *p* = 0.002, Student’s *t*-test). The percentage of CD8^+^ T lymphocytes was also higher in tumors from mice treated with PEK vaccine plus anti-PD-L1 Ab than in tumors from mice treated with PEK vaccine plus PBS (*: *p* = 0.002, Student’s *t*-test). (**D**) Representative figures of tumor-infiltrating CD80^+^ CD11c^+^ and CD86^+^ CD11c^+^ DCs from mice on day 28 after tumor challenge in various groups as determined by flow cytometry. (**E**) Percentages of CD80^+^ CD11c^+^ and CD86^+^ CD11c^+^ DCs from mice on day 28 after tumor challenge in various groups, as determined by flow cytometry (*n* = 5 per group, mean ± SEM). The CD80 maturation marker of CD11c^+^ DCs significantly increased in tumors from mice treated with the PEK protein vaccine plus anti-PD-L1 Ab compared with tumors from mice treated with the PEK vaccine plus PBS (*: *p* = 0.002, Student’s *t*-test). The CD86 maturation marker of CD11c^+^ DCs also significantly increased in tumors from mice treated with the PEK protein vaccine plus anti-PD-L1 Ab compared with tumors from mice treated with the PEK vaccine plus PBS (*: *p* < 0.001, Student’s *t*-test). (**F**) Representative figures depicting tumor-infiltrating CD80^+^ F4/80^+^ and CD86^+^ F4/80^+^ macrophages from mice on day 28 after tumor challenge in various groups, as determined by flow cytometry. (**G**) Percentages of tumor-infiltrating CD80^+^ F4/80^+^ and CD86^+^ F4/80^+^ macrophages from mice on day 28 after tumor challenge in various groups, as determined by flow cytometry (*n* = 5 per group, mean ± SEM). The percentage of CD80-expressing F4/80^+^ macrophages was significantly higher in tumors from mice treated with the PEK protein vaccine plus anti-PD-L1 Ab than in tumors from mice treated with the PEK vaccine plus PBS (*: *p* < 0.001, Student’s *t*-test). The percentage of CD86-expressing F4/80^+^ macrophages was significantly higher in tumors from mice treated with the PEK protein vaccine plus anti-PD-L1 Ab than in tumors from mice treated with the PEK vaccine plus PBS (*: *p* < 0.001, Student’s *t*-test). All experiments were performed independently in triplicate.

**Figure 6 cancers-11-01400-f006:**
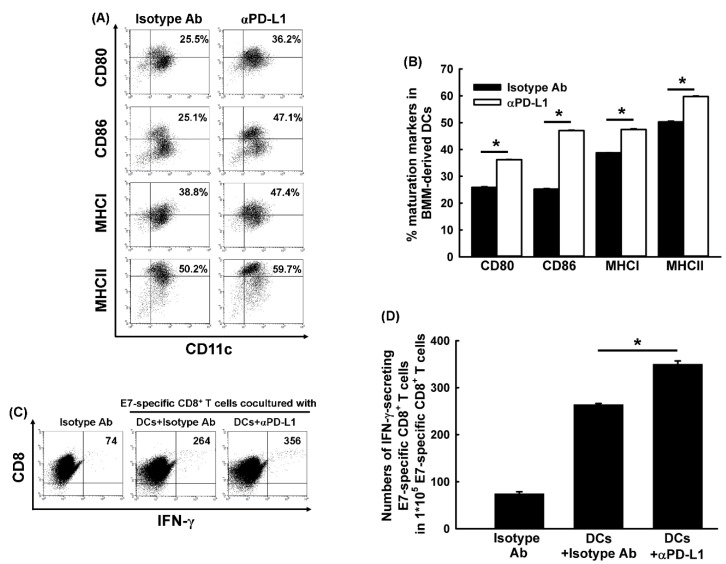
Flow cytometric analysis of the effect of PD-L1 blockade on the expression of various surface markers of BMM-derived DCs and the activation of antigen-specific CD8^+^ cytotoxic T cells mediated by BMM-derived DCs. (**A**) Representative percentages of BMM-derived CD80^+^ CD11c^+^, CD86^+^ CD11c^+^, MHC I^+^ CD11c^+^, and MHC II^+^ CD11c^+^ DCs treated with or without anti-PD-L1 Ab. (**B**) The expression of various surface markers on BMM-derived CD11c^+^ DCs treated with or without anti-PD-L1 Ab. Mice treated with anti-PD-L1 Ab had significantly higher percentages of mature CD11c^+^ DCs than mice treated with the isotype Ab (*: *p* < 0.001, Student’s *t*-test). (**C**) Representative figures depicting IFN-γ secretion by E7-specific CD8^+^ cytotoxic T-cell line co-cultured with E7 peptide-pulsed BMM-derived DCs treated with or without anti-PD-L1 Ab. (**D**) Bar figures depict the number of E7-specific IFN-γ-secreting CD8^+^ cytotoxic T cells in various treatment groups. The number of E7-specific IFN-γ-secreting CD8^+^ T cells was highest in BMM-derived DCs from mice pretreated with anti-PD-L1 Ab (*: *p* < 0.001, Student’s *t*-test). All experiments were performed independently in duplicate (mean ± SEM).

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
