# Peer review of "Blockade of PD-L1 Enhances Cancer Immunotherapy by Regulating Dendritic Cell Maturation and Macrophage Polarization"

_cancers, 2019, doi:10.3390/cancers11091400_

Round 1

Reviewer 1 Report

This is an excellent study demonstrating that anti-PD-L1 antibody therapy enhances the immunotherapeutic effects of a HPV16 E6/E7 vaccine.  The main conclusion of the study is that PD-L1 which is present on the surface of tumor cells and antigen presenting cells performs its therapeutic effect principally by maturing dendritic APC in the tumor microenvironment and tumor draining lymph nodes and bone marrow.  This reviewer has a few comments for the authors to help improve the manuscript, including:

 This study relies heavily on that published in reference #31. (Lin et al).  In the discussion, it would be very helpful to distinguish the findings of the present work with those of Lin et al. In Figure 8, it is recommended that sections (D) and (G) be presented in the same way as section (A) which is easier to read and for consistency. The data presented regarding the maturation of DC (section 2.7) in the tumor microenvironment (TME)is not that dramatic (13.61 vs 9.97%).  Would the authors think that these results might be greater if DC's were monitored in tumor draining lymph nodes or bone marrow compared to the TME?  Most studies investigating synergisms between blockade of the PD-1/PD-L1 axis have shown that innate immune agonists such as TLR agonists, the Sting pathway, and oncolytic viruses provide the best effects, do the authors believe that they are seeing an extension of these prior results, or a new finding that involves adaptive immunity?  This would be a great discussion point in the discussion. Have the authors considered testing the activity of anti-PD-L1 on tumor cells vs DC maturation by using transfer experiments of isolated treated DC's into tumor bearing mice as further proof of their conclusions?

Reviewer 2 Report

SUMMARY:
Your manuscript „Blockade of PD-L1 Enhances Cancer Immunotherapy by Regulating Dendritic Cell Maturation and Macrophage Polarization” shows a coherent proof that a combinatorial therapy of an immune checkpoint inhibitor and a tumor vaccine has great therapeutical potential in future cancer treatment. Furthermore, your work underlines the pivotal role of the expression of PD-L1 in the tumor environment (DCs, macrophages) in the context of immune checkpoint inhibition, which might have even more importance than the PD-L1 expression on tumor cells itself. Both hypothesis are not new and there are even clinical studies published on that topic (some are cited in your manuscript). However, your work provides in vivo and in vitro evidence for both theories. However, using the E7-expressing TC-1 small tumor model in mice is quite artificial, as you won´t find a common immunogenic tumor antigen in every tumor cell of a very aggressive solid tumor like e.g. glioblastoma. Therefore, I would not generalize your final conclusion for cancer therapy.

MAJOR ISSUES:
- The figures are displayed as pictures and not vector graphics, although they contain very small labels. In some cases, you are not able to read the scale bars (e.g. Fig. 2F). Please increase the resolution of the pictures, so that everyone is able to read all legends and scale bars or insert a vector graphic instead.

- A graphic of the experiment design would be useful to understand much faster what you actually did. Maybe you can combine some plots in a multidimensional plot and insert a schematic graph of your experiment design instead.

MINOR ISSUES:

- Please add the university/hospital and the name of the town / country for the first two affiliations

Round 2

Reviewer 2 Report

No further changes required